# A Fast Post-Training Pruning Framework for Transformers

**Woosuk Kwon**[*]
UC Berkeley
woosuk.kwon@berkeley.edu

**Sehoon Kim**[*]
UC Berkeley
sehoonkim@berkeley.edu

**Michael W. Mahoney**
UC Berkeley, ICSI, & LBNL
mahoneymw@berkeley.edu

**Joseph Hassoun**
Samsung Semiconductor, Inc.
j.hassoun@samsung.com

**Kurt Keutzer**
UC Berkeley
keutzer@berkeley.edu

**Amir Gholami**
UC Berkeley
amirgh@berkeley.edu

## Abstract

Pruning is an effective way to reduce the huge inference cost of Transformer models. However, prior work on pruning Transformers requires retraining the models. This can add high training cost and high complexity to model deployment, making it difficult to use in many practical situations. To address this, we propose a fast post-training pruning framework for Transformers that does not require any retraining. Given a resource constraint and a sample dataset, our framework automatically prunes the Transformer model using structured sparsity methods. To retain high accuracy without retraining, we introduce three novel techniques: (i) a lightweight mask search algorithm that finds which heads and filters to prune based on the Fisher information; (ii) mask rearrangement that complements the search algorithm; and (iii) mask tuning that reconstructs the output activations for each layer. We apply our method to BERT$_{\text{BASE}}$ and DistilBERT, and we evaluate its effectiveness on GLUE and SQuAD benchmarks. Our framework achieves up to $2.0\times$ reduction in FLOPs and $1.56\times$ speedup in inference latency, while maintaining $< 1\%$ loss in accuracy. Importantly, our framework prunes Transformers in less than 3 minutes on a single GPU, which is over two orders of magnitude faster than existing pruning approaches that retrain the models.[1]

## 1 Introduction

In recent years, Transformer [76] has become a *de facto* standard model architecture in Natural Language Processing [4, 12, 46], and it is becoming common in many domains including Computer Vision [14, 48, 75] and Speech Recognition [2, 7, 26]. However, efficient deployment of Transformer architectures has been challenging due to their large model size and high inference latency. As a promising way to tackle this challenge, structured pruning of Transformers has been widely studied.

While prior work on pruning Transformers substantially reduces the inference time, it is often difficult to use in practice for several reasons. First, previous approaches require retraining the pruned model and/or jointly learning the pruning configurations during training. This increases the training time by up to $10\times$ [38, 88], adding significant computational overhead. Second, previous methods add many moving parts to the model deployment process. That is, the pruning pipelines are often complex and require additional hyperparameter tuning. Such techniques demand significant engineering efforts for implementation and debugging, which impedes their adoption in production pipelines. Third,

---

[*]Equal contribution.

[1]Our code is publicly available at https://github.com/WoosukKwon/retraining-free-pruning

36th Conference on Neural Information Processing Systems (NeurIPS 2022).

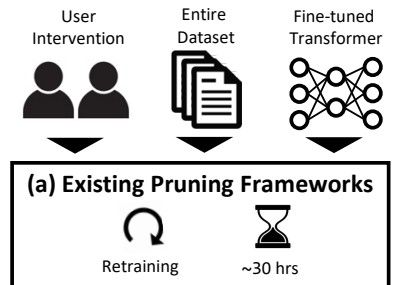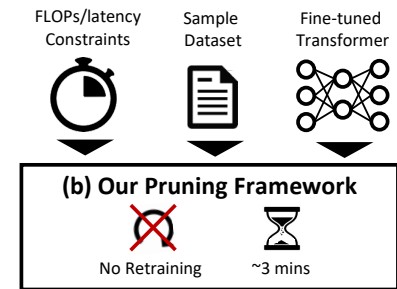

Figure 1: (a) Prior pruning frameworks require additional training on the entire training set and involve user intervention for hyperparameter tuning. This complicates the pruning process and requires a large amount of time (e.g., ∼30 hours). (b) Our pruning framework does not require retraining. It outputs pruned Transformer models satisfying the FLOPs/latency constraints within considerably less time (e.g., ∼3 minutes), without user intervention.

these previous methods do not directly adapt to the users' constraints. They either rely on vague regularization hyperparameters to control the model sparsity or use fixed model architectures selected independently of the user settings. This can result in sub-optimally pruned models that are not tailored to the users' constraints and hardware.

To address the above limitations, we propose a fast *post-training pruning* framework for Transformers that does not require any retraining of the models. As illustrated in Figure 1, our framework takes as input a Transformer model, a sample dataset, and a FLOPs/latency constraint. It then outputs a pruned Transformer model that can be deployed immediately. By avoiding expensive retraining, the end-to-end pruning pipeline can be extremely fast and simplified, which typically takes a few minutes without any user interventions that complicate the whole process.

Indeed, post-training compression has been widely studied for quantization, and it gained considerable attention in both academia and industry [3, 27, 97]. Although quantization-aware training methods achieve higher compression rates in general, post-training quantization (PTQ) has often been more preferred in practice due to its retraining-free advantages. Importantly, PTQ allows quantization to happen seamlessly at the model deployment time using the tools such as TensorRT [56], TFLite [19], and OpenVINO [29]. Similar to the PTQ methods, our framework provides an out-of-the-box tool that enables pruning of Transformers without engineering efforts.

Our contributions can be summarized as follow:

- We propose a novel post-training pruning framework for Transformers that does not require model retraining. To retain accuracy without retraining, our framework consists of three stages: (i) the *mask search* process guided by the Fisher information matrix to select which heads/filters to prune (Section 4.1); (ii) the *mask rearrangement* process that reselects the heads/filters to prune by capturing intra-layer interactions (Section 4.2); and (iii) the *mask tuning* process that adjusts the mask variables to ensure that the output signal is recovered for each layer (Section 4.3).

- We extensively test our framework by applying it to BERT$_{\text{BASE}}$ and DistilBERT on GLUE and SQuAD tasks (Section 5.2). Within 1% of accuracy drop, our framework reduces 30–50% of the original FLOPs (Figure 5), resulting in up to 1.56× speedup on an NVIDIA V100 GPU (Table 1).

- We show that our method achieves comparable or even better FLOPs-accuracy trade-off than prior structured pruning methods *without* retraining (Section 5.3, Figure 6). Our end-to-end pruning pipeline finishes in only 39 and 135 seconds on average for GLUE and SQuAD (Section 5.4, Table 5), which is over 100× faster than the retraining-based methods.

## 2    Related Work

**Efficient Transformers.** In order to improve the inference speed and reduce the memory footprint of Transformers, multiple different approaches have been proposed. These can be broadly categorized as follows: (i) efficient architecture design [28, 35, 39, 72, 81, 87]; (ii) hardware-software co-design [21, 22, 73, 80]; (iii) knowledge distillation [31, 63, 71, 82]; (iv) quantization [33, 66, 95, 96]; (v) neural architecture search [6, 67, 68, 79, 90, 93]; and (vi) pruning. In this paper, we focus on pruning and briefly discuss the related works.

**Transformers Pruning.** Pruning has been a promising way to remove unimportant weights in neural networks. Pruning can be largely categorized into unstructured and structured pruning. For unstructured pruning, magnitude-based [18], first-order [64], and second-order [36] pruning methods, and the lottery ticket hypothesis [9, 10, 16, 58] have been explored for Transformers. While these methods can substantially compress the model size, commodity hardware such as GPUs can hardly take advantage of the unstructured sparse patterns for model inference speedup.

For this reason, a number of structured pruning methods have been introduced to remove coarse-grained sets of parameters in Transformers. For example, to prune structured sets of parameters in weight matrices, low-rank factorization [83], block-wise sparsity [43], and tile-wise sparsity [20] were studied. Furthermore, as more coarse-grained methods, attention head pruning [51, 77] and layer dropping [15, 62] have been popularly used. Taking a step further, recent approaches [8, 32, 38, 44, 47, 88, 92] have explored jointly pruning Transformers with different pruning granularity and principles, maximizing the model efficiency in every dimension. Orthogonally, another thread of work [15, 25, 49, 89, 98] has shown that Transformers can be dynamically pruned at inference time.

Unfortunately, while the structured pruning methods can achieve high compression rates and speedups, they are often difficult to use in practice. One reason for this is the high computational cost of additional training during or after pruning, which can be up to $10\times$ [38, 88] compared to that of the original model training. Another reason is the high complexity of the pruning pipelines [25, 38, 47, 92], where each pruning stage often requires rewriting the training code and introduces additional hyperparameters to tune.

**Post-training Model Compression.** Post-training compression methods have been widely studied in quantization. These methods, categorized as post-training quantization (PTQ), perform quantization without any retraining, thereby avoiding the additional training cost and user intervention. Multiple PTQ techniques have been proposed to effectively mitigate the accuracy degradation without retraining [3, 27, 54, 97].

Although not as much as for quantization, post-training schemes have also been explored for unstructured [17, 40, 53] and structured pruning of CNNs. For structured pruning, [34, 70, 94] proposed ways to group and merge similar neurons in a CNN. However, we find it difficult to extend those techniques to pruning Transformers because they require the model to have a repeating structure of a linear layer and an element-wise nonlinearity, which is not the case for the multi-head attention layers of Transformers. Even for the feed-forward network layers of Transformers, [34, 70] can hardly be used because they rely on a certain characteristic of ReLU while many Transformers [4, 12, 91] use GELU [24] instead of ReLU.

Motivated by the fact that the existing post-training CNN pruning techniques cannot be applied to Transformers, in this paper we propose a novel post-training pruning method with a focus on Transformers. However, we would like to note that the underlying principles in our approach are general enough to be extended to pruning other types of model architectures such as CNNs.

## 3 Overview

### 3.1 Background

**Transformer Architecture.** In this paper, we focus on the pruning of encoder-based Transformer [76] models, especially the BERT [12] architecture family. BERT is a stack of homogeneous Transformer encoder blocks, each of which consists of a multi-head attention (MHA) layer followed by a point-wise Feed-Forward Network (FFN) layer. Specifically, an MHA layer consists of $H$ independently parameterized attention heads:

$$\text{MHA}(\mathbf{x}) = \sum_{i=1}^{H} \text{Att}_i(\mathbf{x}), \quad \mathbf{x}_{\text{MHA}} = \text{LayerNorm}\big(\mathbf{x} + \text{MHA}(\mathbf{x})\big),$$

where Att is a dot product attention head, and x is the input sequence. The output of the MHA layer is then fed into the FFN layer, which consists of $N$ filters:

$$\text{FFN}(\mathbf{x}) = \Big(\sum_{i=1}^{N} \mathbf{W}_{:,i}^{(2)} \sigma(\mathbf{W}_{i,:}^{(1)}\mathbf{x} + b_i^{(1)})\Big) + b^{(2)}, \quad \mathbf{x}_{\text{out}} = \text{LayerNorm}\big(\mathbf{x}_{\text{MHA}} + \text{FFN}(\mathbf{x}_{\text{MHA}})\big),$$

where $\mathbf{W}^{(1)}, \mathbf{W}^{(2)}, b^{(1)}$ and $b^{(2)}$ are the FFN parameters, and $\sigma$ is the activation function, typically GELU [24]. Note that $(H, N)$ is (12, 3072) for BERT$_{\text{BASE}}$, and (16, 4096) for BERT$_{\text{LARGE}}$. We also denote $L$ as the number of Transformer layers (e.g., 12 for BERT$_{\text{BASE}}$).

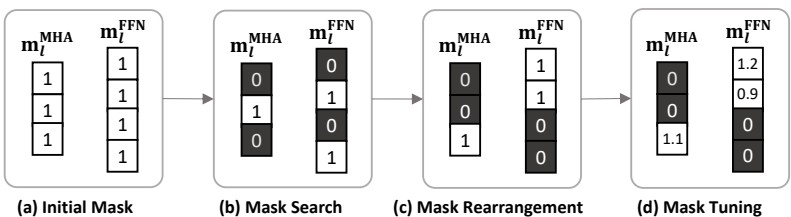

Figure 2: Overview of our pruning framework. (a) The mask variables are initialized as 1. Then they undergo the three-stage pipeline of (b) mask search (Section 4.1), (c) rearrangement (Section 4.2), and (d) tuning (Section 4.3).

**Granularity of Pruning.** Our framework considers the structured pruning of both heads in MHA and filters in FFN layers. We do not prune the embedding and the final classifier, as computation of those layers takes a negligible portion of the total inference latency. Since our pruning framework always produces a smaller dense architecture, the model can be readily accelerated without the need of specialized hardware logic, which is often required for unstructured sparsity to gain latency speedup.

**Notations.** We pose the pruning problem as finding a sparse mask for the heads and filters. To formalize this, we introduce mask variables associated with the outputs of heads and filters:

$$\text{MHA}(\text{x}; \text{m}_l^{\text{MHA}}) = \sum_{i=1}^{H} m_{l,i}^{\text{MHA}} \circ \text{Att}_i(\text{x}),$$

$$\text{FFN}(\text{x}; \text{m}_l^{\text{FFN}}) = \Big( \sum_{i=1}^{N} m_{l,i}^{\text{FFN}} \circ \text{W}_{:,i}^{(2)} \sigma(\text{W}_{i,:}^{(1)}\text{x} + b_i^{(1)}) \Big) + b^{(2)},$$

where $\text{m}_l^{\text{MHA}} \in \mathbb{R}^H$ and $\text{m}_l^{\text{FFN}} \in \mathbb{R}^N$ are the mask variables for MHA and FFN in the $l$-th layer, respectively, and $m_{l,i}^{\text{MHA}}$ and $m_{l,i}^{\text{FFN}}$ are their $i$-th elements. Furthermore, $\circ$ denotes the Hadamard product. Originally, the mask variables are all initialized to 1, which does not change the model outputs. After pruning, the mask variables become zero or any nonzero values, affecting the model accuracy and sparsity. Especially, setting $m_{l,i}^{\text{MHA}}$ and $m_{l,i}^{\text{MHA}}$ as zero is equivalent to pruning the $i$-th head and filter, respectively.

Overall, there are $LH$ head mask variables and $LN$ filter mask variables, summing up to $L(H + N)$ number of total mask variables in a Transformer model. To simplify notations, we additionally define $\text{m}^{\text{MHA}} \in \mathbb{R}^{LH}$, $\text{m}^{\text{FFN}} \in \mathbb{R}^{LN}$, and $\text{m} \in \mathbb{R}^{L(H+N)}$ as flattened vectors of the head, filter, and total mask variables, respectively, across all layers. In what follows, we discuss how to find the optimal sparse masks under a given cost constraint and how to adjust their values to recover accuracy.

## 3.2 Framework Overview

Figure 1(b) and Figure 2 illustrate the overview of our framework.

**Inputs.** Our framework has 3 inputs: a Transformer model; a sample dataset; and a resource constraint. The input Transformer model should contain weights fine-tuned for a downstream task. The sample dataset is a small partition of the training dataset (typically 1–2K examples) for the downstream task. The resource constraint can be given either as the number of floating point operations (FLOPs) or as an actual latency on target hardware. In the later case, we further assume that a latency lookup table for the target hardware is provided.

**Compression Pipeline.** As illustrated in Figure 2, our framework consists of 3 stages: Fisher-based mask search; Fisher-based mask rearrangement; and mask tuning. During the *Fisher-based mask search* stage (Section 4.1), we search for a binary mask applied to the heads and filters based on the Fisher information of the mask variables. Intuitively, the mask variables with relatively higher Fisher information are considered more important, and they should be less likely to be pruned [41, 45, 52]. As finding the optimal mask that minimizes the Fisher information loss is intractable due to the large size of the full Fisher matrix, we propose a lightweight search algorithm that finds the optimal mask under reasonable approximations. Second, in the *Fisher-based mask rearrangement* stage (Section 4.2), the framework adjusts the searched mask patterns to better take into account the intra-layer interactions between the mask variables. Lastly, in the *mask tuning* stage (Section 4.3), the framework tunes the nonzero mask variables to recover the accuracy drop by reconstructing the layer-wise output signal.

## 4  Methodology

The pruning problem can be seen as finding an optimal mask under a sparsity constraint. However, without retraining, the problem becomes intractable. To address this, we decompose Transformer pruning into three sub-problems, each of which can be efficiently solved; the first two stages of our pipeline address the problems of finding an optimal *binary* mask, and the last stage further optimizes it into a *real-valued* mask.

Note that the number of the mask variables is much less than the number of the parameters in a Transformer (e.g., 37K vs. 110M in case of BERT$_{\text{BASE}}$). This allows the framework to use only a small number of examples without overfitting to the sample dataset, and thus to be extremely faster than the retraining-based pruning methods which typically use the entire dataset. As the framework keeps the model "as is" and only decides the mask variables, we henceforth regard the model parameters as constants and consider the mask variables as the only parameters for our pruning problem.

**Problem Formulation.** We formulate Transformer pruning as a constrained optimization problem on the mask m:

$$\arg\min_{\mathbf{m}} \mathcal{L}(\mathbf{m}) \quad \text{s.t.} \quad \text{Cost}(\mathbf{m}) \leq C \tag{1}$$

where $\mathcal{L}$ denotes the loss function, Cost is the FLOPs/latency of the architecture pruned by the mask, and $C$ is the given FLOPs/latency constraint. Unfortunately, such a problem is generally intractable as Cost is usually a function of $l_0$-norm of the mask m, which is non-differentiable. Thus, in what follows, we introduce several assumptions and approximations to simplify the problem.

We start by approximating the loss function using the second-order Taylor expansion around the initial mask $\mathbb{1}$:

$$\mathcal{L}(\mathbf{m}) \approx \mathcal{L}(\mathbb{1}) - \mathbf{g}^{\mathsf{T}}(\mathbb{1} - \mathbf{m}) + \frac{1}{2}(\mathbb{1} - \mathbf{m})^{\mathsf{T}}\mathbf{H}(\mathbb{1} - \mathbf{m}) \tag{2}$$

$$\approx \mathcal{L}(\mathbb{1}) + \frac{1}{2}(\mathbb{1} - \mathbf{m})^{\mathsf{T}}\mathbf{H}(\mathbb{1} - \mathbf{m}), \tag{3}$$

where $\mathbf{g} = \mathbb{E}[\frac{\partial}{\partial \mathbf{m}}\mathcal{L}(\mathbb{1})]$ and $\mathbf{H} = \mathbb{E}[\frac{\partial^2}{\partial \mathbf{m}^2}\mathcal{L}(\mathbb{1})]$. Eq. 3 is deduced from an assumption that the model has converged to a local minima, where the gradient term is close to 0 [41]. As $\mathcal{L}(\mathbb{1})$ is a constant, we can rewrite the optimization objective as follows:

$$\arg\min_{\mathbf{m}} \mathcal{L}(\mathbf{m}) \approx \arg\min_{\mathbf{m}}(\mathbb{1} - \mathbf{m})^{\mathsf{T}}\mathbf{H}(\mathbb{1} - \mathbf{m}). \tag{4}$$

Eq. 4 shows that the optimal mask is determined by the Hessian of the loss with respect to the mask variables. Since forming the exact Hessian matrix explicitly is infeasible, we approximate the Hessian H with the (empirical) Fisher information matrix $\mathcal{I}$ of the mask variables:

$$\mathcal{I} := \frac{1}{|\mathcal{D}|} \sum_{(x,y) \in \mathcal{D}} \left(\frac{\partial}{\partial \mathbf{m}}\mathcal{L}(x, y; \mathbb{1})\right)\left(\frac{\partial}{\partial \mathbf{m}}\mathcal{L}(x, y; \mathbb{1})\right)^{\mathsf{T}}, \tag{5}$$

where $\mathcal{D}$ is the sample dataset and $(x, y)$ is a tuple of an input example and its label.

### 4.1  Fisher-based Mask Search

**Diagonal Approximation of the Fisher Information Matrix.** It is intractable to solve the optimization objective in Eq. 4 using the full Fisher information matrix $\mathcal{I}$. Thus, we first make a simple assumption that $\mathcal{I}$ is *diagonal*. This further simplifies Eq. 4 as follows:

$$\arg\min_{\mathbf{m}} \mathcal{L}(\mathbf{m}) \approx \arg\min_{\mathbf{m}} \sum_i (1 - m_i)^2 \mathcal{I}_{ii}, \tag{6}$$

Since we restrict the possible mask values to either 0 or 1, the following can be derived from Eq. 6:

$$\arg\min_{\mathbf{m}} \mathcal{L}(\mathbf{m}) \approx \arg\min_{\mathbf{m}} \sum_{i \in Z(\mathbf{m})} \mathcal{I}_{ii} \quad \text{where} \quad Z(\mathbf{m}) := \{i \mid m_i = 0\}. \tag{7}$$

We can interpret each diagonal element of $\mathcal{I}$ as the *importance score* of the head/filter associated with the mask variable, and Eq. 7 as a process of minimizing the total importance scores of the pruned heads and filters. Such an importance score has also been introduced in [52, 74] to guide pruning.

**Algorithm 1** Mask Search with a FLOPs Constraint

---

**Input:** FLOPs constraint $C$, diagonal Fisher information matrix $\mathcal{I}$

1: **for** $n = 0$ **to** $LH$ **do**                                  ▷ # remaining heads
2:    $k_1 = LH - n$                                             ▷ # heads to prune
3:    HI = indicies of $k_1$ least important heads
4:    $f = \lfloor (C - n\text{F}_{\text{head}})/\text{F}_{\text{filter}} \rfloor$                    ▷ # remaining filters
5:    $k_2 = LN - f$                                             ▷ # filters to prune
6:    FI = indicies of $k_2$ least important filters
7:    $S[n] = \sum_{i \in \text{HI} \cup \text{FI}} \mathcal{I}_{ii}$
8:    $R[n] = (\text{HI}, \text{FI})$
9: **end for**
10: $n^* = \arg\min_n S[n]$                                   ▷ optimal # remaining heads
11: $\text{HI}^*, \text{FI}^* = R[n^*]$                          ▷ indicies of heads/filters to prune
12: Initialize $\text{m}^{\text{MHA}}$ and $\text{m}^{\text{FFN}}$ as $\mathbb{1}$
13: $\text{m}^{\text{MHA}}[\text{HI}^*] = 0$                         ▷ prune the selected heads
14: $\text{m}^{\text{FFN}}[\text{FI}^*] = 0$                          ▷ prune the selected filters
**Output:** $\text{m}^* = (\text{m}^{\text{MHA}}, \text{m}^{\text{FFN}})$

---

**Solving FLOPs-constrained Problem.** We need to solve Eq. 7 given a cost constraint. For a given target FLOPs cost, denoted by C, we can formulate the binary mask search problem as follows:

$$\arg\min_{\text{m}} \sum_{i \in Z(\text{m})} \mathcal{I}_{ii} \quad \text{s.t.} \quad \text{F}_{\text{head}} ||\text{m}^{\text{MHA}}||_0 + \text{F}_{\text{filter}} ||\text{m}^{\text{FFN}}||_0 \leq C, \tag{8}$$

where $\text{F}_{\text{head}} \in \mathbb{R}$ and $\text{F}_{\text{filter}} \in \mathbb{R}$ are the FLOPs for computing a head and a filter, respectively. Note that the number of FLOPs of a head/filter is constant across all layers. While such an optimization problem can be generally solved by a knapsack algorithm [1, 65], the following observations allow a faster polynomial-time solution: (1) having more heads and filters unpruned always optimizes Eq. 8 since the diagonal elements of $\mathcal{I}$ are non-negative; and (2) if a certain number of heads needs to be pruned, they should be the ones with the lowest importance scores because each head accounts for the same amount of FLOPs. The same statement also holds for pruning filters. The two observations lead to our mask search algorithm described in Algorithm 1.

Algorithm 1 partitions the solution space by the total number of remaining heads in the pruned architecture ($n$ in line 1). For each $n$, the number of remaining filters should be the largest possible number that satisfies the cost constraint by observation (1), which can be described as $f$ in line 4. Then by observation (2), the heads/filters with the lowest important scores are selected to be pruned. Therefore, $S[n]$ is a solution of Eq. 8 under additional constraint of fixing the number of remaining heads to be $n$. When the loop terminates, the output is the mask that minimizes $S[n]$ across all possible $n$ (line 10 and 11). In Section A.1, we prove that the output mask $\text{m}^*$ of Algorithm 1 is optimal. That is, any other mask m satisfying the given FLOPs constraint will result in a higher loss:

$$\sum_{i \in Z(\text{m}^*)} \mathcal{I}_{ii} \leq \sum_{i \in Z(\text{m})} \mathcal{I}_{ii}. \tag{9}$$

**Solving Latency-constrained Problem.** If the cost constraint is given in terms of latency on target hardware, we have a new optimization problem with a different cost constraint than Eq. 8:

$$\arg\min_{\text{m}} \sum_{i \in Z(\text{m})} \mathcal{I} \quad \text{s.t.} \quad \sum_{l=1}^{L} \text{LAT}(\text{m}_l^{\text{MHA}}) + \sum_{l=1}^{L} \text{LAT}(\text{m}_l^{\text{FFN}}) \leq C, \tag{10}$$

where the function LAT indicates the latency of a MHA/FFN layer after pruning. We assume that a latency lookup table on the target hardware is provided so that evaluating LAT takes negligible time.

Unfortunately, the latency constraint makes the problem more challenging as directly applying Algorithm 1 is no longer possible. This is because LAT is *not* linear to the number of remaining heads or filters after pruning [59], as shown in Figure 3 (Left). We can interpret this as follows: (1) with a sufficient number of heads/filters in a layer, the hardware resources such as parallel cores can be fully utilized, resulting in latency roughly proportional to the number of heads/filters; and (2) otherwise, the hardware is underutilized and a constant overhead dominates the latency [37, 50]. Thus, pruning more heads/filters below a certain threshold does not translate into actual speedup.

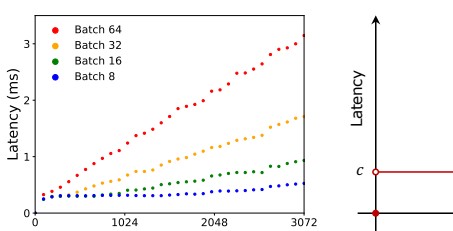
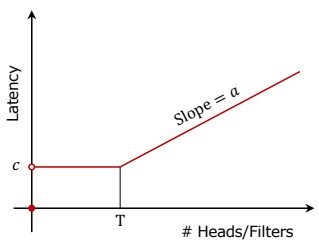
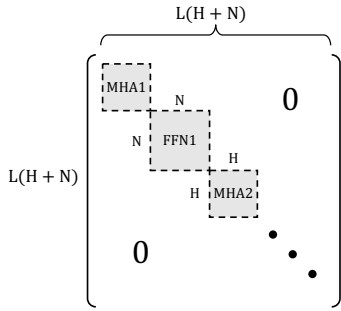

Figure 3: (Left) Real latency of a single FFN layer with different numbers of remaining filters. (Right) Schematic plot for the approximated latency as a piece-wise linear function.

Figure 4: Illustration of the block diagonal Fisher matrix.

Based on the above analysis, we approximate LAT as a piece-wise linear function as in Figure 3 (Right) such that $LAT(m_l)$ is 0 if $||m_l||_0 = 0$, $c$ if $0 < ||m_l||_0 \leq T$, and $a(||m_l||_0 - T) + c$ if $||m_l||_0 > T$, where $c \in \mathbb{R}$ is the constant overhead, $T \in \mathbb{N}$ is the threshold number of heads/filters that the latency starts to become linear, and $a \in \mathbb{R}$ is the slope of the linear part. This can be easily obtained by fitting the actual latency data in the lookup table with the minimum mean squared error.

The piece-wise linear approximation of LAT allows us to extend Algorithm 1 to the setting with latency constraints. The core idea is to separately consider the constant part and the linear part of LAT; after handling the constant part, we can apply the Algorithm 1 to the linear part. The detailed modification to Algorithm 1 is described in Section A.2.

### 4.2 Fisher-based Mask Rearrangement

**Block Diagonal Approximation of the Fisher Information Matrix.** Although it simplifies the problem, the diagonal assumption in Section 4.1 alone might not find the best solution, as it does not take into account the interactions between different mask variables. For example, if there are two attention heads playing a similar role in a layer, pruning only one of them might not affect the model accuracy. However, when both of them are pruned, the model accuracy can be significantly degraded. Such interactions are captured by the non-diagonal elements of the Fisher information matrix, which were ignored in the previous stage. Thus, we can better consider the interactions in our pruning problem by using a *block diagonal* approximation to the Fisher matrix, where a block corresponds to a MHA layer or a FFN layer as illustrated in Figure 4.

However, the block diagonal approximation results in an intractable optimization problem over the binary mask. To alleviate this, we use the results from the previous stage to *warm start* the optimization problem. First, we constrain the number of heads/filters to prune for each layer to be the same as the binary mask we obtained in the first stage. In other words, given the mask $m^*$ obtained in Section 4.1, we constrain $||m_l||_0$ to be equal to $||m_l^*||_0$ for each layer $l$. Second, we use the mask $m^*$ as the starting point of the greedy search to solve the new optimization problem.

Given the two assumptions that (i) there is no interaction between the mask variables in different layers (i.e., the block diagonal approximation), and (ii) the number of heads/filters to prune are pre-determined for each layer (i.e., warm-start), Eq. 4 breaks down to a set of *layer-wise* optimization problems, as follows based on the derivation in Section A.3:

$$\hat{m}_l = \underset{m_l}{\arg \min}(\mathbb{1} - m_l)^\intercal \mathcal{I}_l(\mathbb{1} - m_l), \tag{11}$$

where $\mathcal{I}_l$ is the $l$-th diagonal block of $\mathcal{I}$. We approximately solve this problem with a greedy algorithm. After initializing the mask $m_l$ as $m_l^*$ (i.e., warm-start), we pick for every round a pruned head (or filter) with the highest Fisher information and exchange it with an unpruned head (or filter) in the current mask if that can further optimize Eq. 11. After every pruned head/filter goes through one round, we obtain an approximate solution to Eq. 11.

Because this process does not change the number of heads/filters in each layer, the obtained mask $\hat{m}_l$ results in the same FLOPs/latency as that of the mask $m_l^*$ searched in Section 4.1. In effect, this process *rearranges* the binary mask variables of each layer to find a better arrangement of pruning locations and capture the intra-layer interactions.

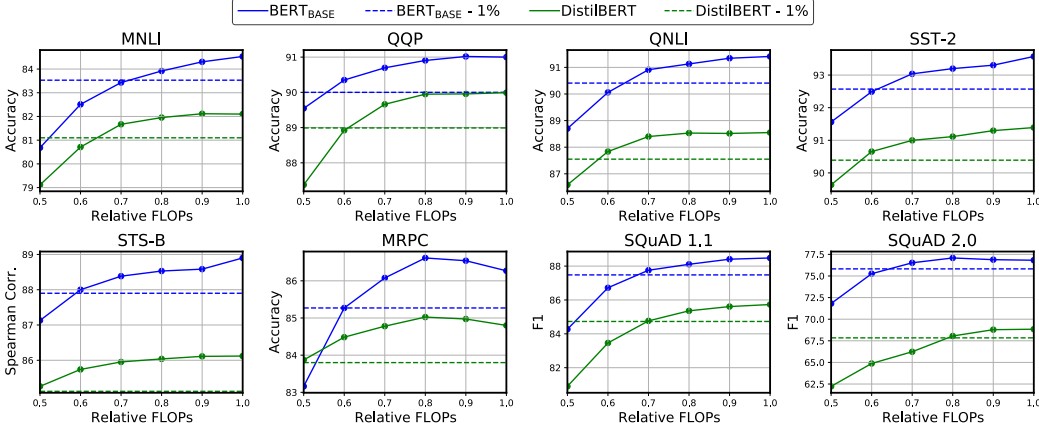

Figure 5: Accuracy of our pruning method applied to BERT$_{\text{BASE}}$ and DistilBERT with different FLOPs constraints. The dashed horizontal lines indicate 1% accuracy drop from the baseline models. Note that these results can be achieved in only 39 and 135 seconds for GLUE and SQuAD benchmarks, respectively, on a single GPU system, as described in Table 5 (in Section A.10).

Table 1: Latency speedup of BERT$_{\text{BASE}}$ on a single NVIDIA V100 GPU with different batch sizes. The latency is measured using PyTorch. We constrain the accuracy degradation to be at most 1% from the baseline accuracy, and we report the largest speedup among those that satisfy the constraint.

| Batch size | MNLI | QQP | QNLI | SST-2 | STS-B | MRPC | SQuAD$_{1.1}$ | SQuAD$_{2.0}$ | Geo. mean |
|---|---|---|---|---|---|---|---|---|---|
| 32 | 1.27× | 1.42× | 1.42× | 1.23× | 1.34× | 1.36× | 1.33× | 1.37× | 1.34× |
| 256 | 1.34× | 1.54× | 1.53× | 1.56× | 1.54× | 1.55× | 1.34× | 1.40× | 1.47× |

### 4.3 Mask Tuning

In the previous two stages, the possible mask values are restricted to either 0 or 1 in order to simplify the search process. In this stage, we further relax this restriction. The *nonzero* variables in the mask m̂ from Section 4.2 are tuned to any real values such that the pruned model recovers its accuracy.

**Layer-wise Reconstruction via Linear Least Squares.** We tune the mask variables toward minimizing the *layer-wise reconstruction error*, similarly to [23]. From the first to the last layer, we reconstruct the output activation of the original model with the remaining heads/filters in the pruned model. This can be formally written as follows:

$$\arg\min_{\text{m}_l} ||\text{x} + \text{layer}(\text{x}; \text{m}_l) - \big(\text{x}' + \text{layer}(\text{x}'; \mathbb{1})\big)||_2^2, \tag{12}$$

where layer is either MHA or FFN, and x and x′ are the inputs to the layer of the pruned model and the original model, respectively. Here we compare the activations after the residual connection. Note that this stage does not incur any change in model FLOPs/latency, as we only tune the nonzero mask variables. We show in Section A.4 that Eq. 12 can be reduced to a *linear least squares* problem of $\arg\min_{\text{m}_l} ||\text{Am}_l - \text{b}||_2^2$, where the matrix A denotes head/filter-wise output activations of the model pruned by the binary mask and the vector b is the difference between the output activations of the two models. Concretely, when there are $T$ tokens in the sample dataset and $D$ is the hidden size of the model, the size of the matrix A is $TD \times H$ for head masks and $TD \times N$ for filter masks.

Due to the large size of the matrix A, naively solving the least squares problem can lead to numerically unstable results. To address this, our framework uses the LSMR solver in CuPy [55] with a regularization hyperparameter (i.e., `damp`). Concretely, we re-parameterize the least squares problem as $\arg\min_{\text{r}_l} ||\text{Ar}_l + \text{A} \cdot \mathbb{1} - \text{b}||_2^2$ where $\text{m}_l = \mathbb{1} + \text{r}_l$, and solve it with the `damp` value fixed to 1. Then, to prevent the case in which the tuned mask rather hurts the accuracy, we restrict the acceptable range of the tuned mask variables to [-10, 10]. When the solver finds a layer mask that exceeds this range, we discard the mask for that layer and stop mask tuning. In our experiments, we find that the aforementioned heuristics make the mask tuning process highly stable across different models, tasks, and seeds. Furthermore, while the use of the heuristics involves the two hyperparameters (i.e., `damp` and the acceptable range), we empirically find that these need not be tuned for different tasks and models. In all of our experiments, we fixed the two hyperparameter values as we mentioned here.

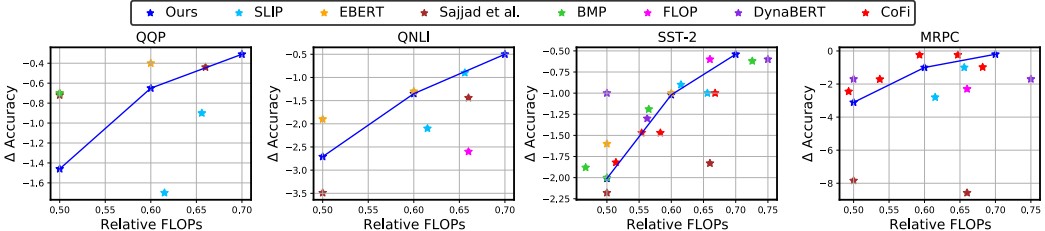

Figure 6: Amount of accuracy degradation from the baseline when pruning BERT$_{BASE}$ using our method and the prior structured pruning methods with different relative FLOPs. Note that our method does not require retraining, whereas all the other methods involve significant retraining overheads as described in Table 2.

## 5 Evaluation

### 5.1 Experimental Setup

Our framework is implemented on top of PyTorch [57] and the HuggingFace Transformers [86] library. We evaluate the effectiveness of our approach using BERT$_{BASE}$ [12] and DistilBERT [63] on GLUE [78] and SQuAD [60, 61] benchmarks. We use 2K examples from the training sets for pruning, and we evaluate the resulting models on the development sets. All of the results are averaged over the runs with 10 different seeds. More details on the experimental setup can be found in Section A.5.

### 5.2 Performance Evaluation

**FLOPs.** Figure 5 shows the accuracy of BERT$_{BASE}$ and DistilBERT with different FLOPs constraints on GLUE and SQuAD datasets. As can be seen in the plots, with only 1% of accuracy drop, BERT$_{BASE}$ achieves 60–70% of the original FLOPs for all tasks. DistilBERT also shows a similar pattern and achieves up to 50% FLOPs reduction (in STS-B and MRPC) even though it is already a compressed architecture. More results using larger sample datasets are provided in Section A.6.

**Latency.** We further measure the latency on real hardware by pruning BERT$_{BASE}$ with latency constraints and deploying the resulting models on an NVIDIA V100 GPU. Table 1 lists the latency speedup with maximum accuracy drop of 1% for GLUE and SQuAD datasets. With batch size of 256, we achieve speedup of $1.47\times$ on average and up to $1.56\times$.

### 5.3 Comparison with the Prior Methods

**FLOPs and Accuracy Comparison.** Here, we compare our method with the prior structured pruning methods for Transformers including Flop [83], SLIP [44], Sajjad et al. [62], DynaBERT [25], EBERT [49], Block Movement Pruning (BMP) [38], and CoFi [88] by the FLOPs-accuracy trade-off of BERT$_{BASE}$ on GLUE tasks. We use the results *without* knowledge distillation and data augmentation reported in each paper. Since the baseline accuracy differs slightly from paper to paper, we compare the amount of the accuracy drop from the baseline instead of the absolute accuracy. The results are plotted as Figure 6. We include the comparison details and full table in Section A.7.

Interestingly, our method exhibits comparable or better results than the prior methods *without* any model retraining and with substantially lower pruning costs. This empirically demonstrates that retraining and a complex pruning pipeline are *not* necessary for moderate level of pruning of Transformers. For high sparsity, we find that our framework *with* retraining works comparably to or better than the prior methods at the same pruning cost (See Section A.8).

**Retraining Cost.** We select DynaBERT [25], EBERT [49], BMP [38], and CoFi [88] that achieve comparably good accuracy in Figure 6, and we systematically analyze their end-to-end retraining costs on MNLI dataset. As shown in Table 2, these methods require $5-33$ hours of retraining. On the other hand, our method finishes in less than a minute, which is $2-3$ orders of magnitude faster. We also highlight that this training latency analysis only accounts for a *single* hyperparameter, and the entire cost should be multiplied by the size of the hyperparameter space. While the prior methods rely on a considerable number of hyperparameters, ours introduce only two hyperparameters (in Section 4.3) which we fix for all experiments. See Section A.9 for more details.

Table 2: Pruning cost comparison between the prior structured pruning methods and ours. We compare the number of training epochs and the end-to-end (E2E) time required for pruning.

|  | # Epochs | E2E time (hr) |
|---|---|---|
| DynaBERT [25] | 4 | 12 |
| EBERT [49] | 6 | 5 |
| BMP [38] | 20 | 17 |
| CoFi [88] | 40 | 33 |
| Ours | **0** | **0.01** |

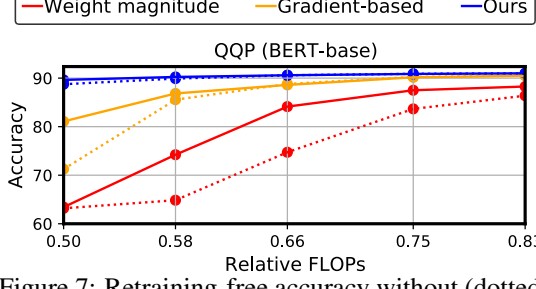

Figure 7: Retraining-free accuracy without (dotted) and with (solid) mask tuning.

Table 3: Ablation of our mask search, rearrangement, and tuning methods, described in Section 4. We use BERT$_{BASE}$ as the baseline model, and we prune it with a 60% FLOPs constraint.

|  | MNLI | QQP | QNLI | SST-2 | STS-B | MRPC | SQuAD$_{1.1}$ | SQuAD$_{2.0}$ | Avg. Diff |
|---|---|---|---|---|---|---|---|---|---|
| Baseline | 84.53 | 91.00 | 91.41 | 93.57 | 88.90 | 86.27 | 88.48 | 76.82 | |
| Mask Search | 81.21 | 89.99 | 88.38 | 92.13 | 87.10 | 83.14 | 82.66 | 71.12 | |
| + Mask Rearrangement | 81.81 | 90.08 | 88.77 | 92.09 | 87.68 | 83.23 | 84.47 | 72.38 | + 0.60 |
| + Mask Tuning | **82.51** | **90.35** | **90.06** | **92.49** | **88.00** | **85.27** | **86.72** | **75.26** | **+ 1.27** |

## 5.4 Discussion

**Ablation Studies.** Table 3 lists an ablation of the mask rearrangement (Section 4.2) and tuning (Section 4.3) stages for pruned BERT$_{BASE}$ with 60% of FLOPs. We find that both stages help recover the baseline accuracy, and that mask tuning is in particular critical, recovering up to 2.88% accuracy.

To further investigate the importance of the mask search and rearrangement, we compare the *retraining-free* performance of the binary masks obtained by our method and other pruning criteria: weight magnitude and the gradient-based method used in DynaBERT. We uniformly pruned the layers using the two criteria with different width multipliers. Figure 7 shows that the two methods significantly degrade the accuracy under the low sparsity regimes. Even with mask tuning, the accuracy is not fully recovered. The results demonstrate that our mask search and re-arrangement are necessary to get optimal binary masks, and that mask tuning is only effective when the binary mask preserves high accuracy. More ablation studies can be found in Section A.11 and Section A.12.

**Time Breakdown.** We break down our pruning pipeline into 4 parts—gradient computation, mask search, rearrangement, and tuning—and we measure the latency for each stage as Table 5 (Section A.10). For GLUE and SQuAD tasks, our framework finishes in 39 and 135 seconds on average.

## 6 Conclusion

In this work, we have proposed a novel post-training pruning framework for Transformers that does not require model retraining. The core of our framework is the three-stage decomposition of the pruning process. It uses a fast Fisher-based mask search algorithm to decide which heads/filters to prune, rearranges the pruned heads/filters, and tunes the mask variables to recover the output signal for each layer. We empirically evaluate our framework using BERT$_{BASE}$ and DistilBERT, where our pruning method achieves up to 50% FLOPs reduction within only 1% accuracy degradation on GLUE and SQuAD datasets. This results in up to $1.56\times$ latency speedup on an NVIDIA V100 GPU. Importantly, our end-to-end pruning pipeline only needs 39 and 135 seconds for GLUE and SQuAD, which is $2-3$ orders of magnitude faster than the prior methods.

## Acknowledgments and Disclosure of Funding

The authors would like to thank Suhong Moon who helped with brainstorming. We also acknowledge gracious support from Google Cloud, Google TRC team, and specifically Jonathan Caton, Prof. David Patterson, and Jing Li. Prof. Keutzer's lab is sponsored by Samsung, Intel corporation, Intel VLAB team, Intel One-API center of excellence, as well as funding through BDD and BAIR. Woosuk Kwon and Sehoon Kim acknowledge the support from Korea Foundation for Advanced Studies. Amir Gholami was supported through funding from Samsung SAIT. Michael W. Mahoney would also like to acknowledge the UC Berkeley CLTC, ARO, NSF, and ONR. Our conclusions do not necessarily reflect the position or the policy of our sponsors, and no official endorsement should be inferred.

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
