# A   Appendix

## A.1   Proof of Equation 9

We prove Eq. 9 by contradiction. Suppose that there exists a mask m such that

$$\sum_{i \in Z(\mathrm{m})} \mathcal{I}_{ii} < \sum_{i \in Z(\mathrm{m}^*)} \mathcal{I}_{ii}, \tag{13}$$

where the mask $\mathrm{m}^*$ is the output of Algorithm 1. Let $h = ||\mathrm{m}^{\mathrm{MHA}}||_0$, i.e. the total number of heads in the architecture pruned by the mask m. Then we construct a new mask $\mathrm{m}'$ as follows:

1. Keep the mask for MHA layers. That is, $\mathrm{m}'^{\mathrm{MHA}} = \mathrm{m}^{\mathrm{MHA}}$.

2. Construct $\mathrm{m}'^{\mathrm{FFN}}$ as in the inner statements of the for loop (line 1 of Algorithm 1). That is, given a mask initialized to $\mathbb{1}$, we zero out the $k$ filter mask variables with the least important scores, where $k = LN - \lfloor (C - f\mathrm{F}_{\mathrm{head}})/\mathrm{F}_{\mathrm{filter}} \rfloor$.

Obviously, the mask $\mathrm{m}'$ satisfies the FLOPs constraint Eq. 8. Moreover, as the mask $\mathrm{m}'$ prunes the least important filters, the following two inequalities hold:

$$\sum_{i \in Z(\mathrm{m}'^{\mathrm{FFN}})} \mathcal{I}_{ii} \leq \sum_{i \in Z(\mathrm{m}^{\mathrm{FFN}})} \mathcal{I}_{ii}, \tag{14}$$

$$\sum_{i \in Z(\mathrm{m}')} \mathcal{I}_{ii} = \sum_{i \in Z(\mathrm{m}'^{\mathrm{MHA}})} \mathcal{I}_{ii} + \sum_{i \in Z(\mathrm{m}'^{\mathrm{FFN}})} \mathcal{I}_{ii} \leq \sum_{i \in Z(\mathrm{m}^{\mathrm{MHA}})} \mathcal{I}_{ii} + \sum_{i \in Z(\mathrm{m}^{\mathrm{FFN}})} \mathcal{I}_{ii} = \sum_{i \in Z(\mathrm{m})} \mathcal{I}_{ii}. \tag{15}$$

Then we construct another mask $\mathrm{m}^\star$ from $\mathrm{m}'$ such that:

1. Keep the mask for FFN layers. That is, $\mathrm{m}^{\star \mathrm{FFN}} = \mathrm{m}'^{\mathrm{FFN}}$.

2. Construct $\mathrm{m}^{\star \mathrm{MHA}}$ by zeroing out $h$ head mask variables with the least important scores from a mask initialized to $\mathbb{1}$.

Due to the observation (2) in Section 4.1, the following inequalities hold:

$$\sum_{i \in Z(\mathrm{m}^{\star \mathrm{MHA}})} \mathcal{I}_{ii} \leq \sum_{i \in Z(\mathrm{m}'^{\mathrm{MHA}})} \mathcal{I}_{ii}, \tag{16}$$

$$\sum_{i \in Z(\mathrm{m}^\star)} \mathcal{I}_{ii} = \sum_{i \in Z(\mathrm{m}^{\star \mathrm{MHA}})} \mathcal{I}_{ii} + \sum_{i \in Z(\mathrm{m}^{\star \mathrm{FFN}})} \mathcal{I}_{ii} \leq \sum_{i \in Z(\mathrm{m}'^{\mathrm{MHA}})} \mathcal{I}_{ii} + \sum_{i \in Z(\mathrm{m}'^{\mathrm{FFN}})} \mathcal{I}_{ii} = \sum_{i \in Z(\mathrm{m}')} \mathcal{I}_{ii}. \tag{17}$$

Essentially, $\mathrm{m}^\star$ is the mask when $n$ (in line 1 of Algorithm 1) is $h$. As Algorithm 1 finds the minimum by iterating different values of $n$, the following inequalities hold:

$$\sum_{i \in Z(\mathrm{m}^*)} \mathcal{I}_{ii} \leq \sum_{i \in Z(\mathrm{m}^\star)} \mathcal{I}_{ii}. \tag{18}$$

Finally, the above inequalities are combined as follows:

$$\sum_{i \in Z(\mathrm{m}^*)} \mathcal{I}_{ii} \leq \sum_{i \in Z(\mathrm{m}^\star)} \mathcal{I}_{ii} \leq \sum_{i \in Z(\mathrm{m}')} \mathcal{I}_{ii} \leq \sum_{i \in Z(\mathrm{m})} \mathcal{I}_{ii}, \tag{19}$$

which contradicts Eq. 13. Thus, the output mask $\mathrm{m}^*$ of Algorithm 1 is the minimizer of Eq. 8.  $\square$

## A.2   Latency-aware Search Algorithm

Algorithm 2 is our proposed algorithm for latency-aware mask search, which extends Algorithm 1. It takes as inputs the given latency constraint, approximated LAT functions for MHA and FFN layers, and the diagonal Fisher information matrix $\mathcal{I}$. Overall, Algorithm 2 has the same structure as Algorithm 1. A notable difference between the two is that Algorithm 2 separately considers the constant part (i.e., when the number of heads/filters is below the threshold $T$) in line 1–5. Another difference is that Algorithm 2 uses $a_{\mathrm{head}}$ and $a_{\mathrm{filter}}$ instead of $\mathrm{F}_{\mathrm{head}}$ and $\mathrm{F}_{\mathrm{filter}}$ in Algorithm 1.

**Algorithm 2** Mask Search with a Latency Constraint

**Input:** Latency constraint $C$, LAT function parameters $(a_{\text{head}}, c_{\text{head}}, T_{\text{head}}), (a_{\text{filter}}, c_{\text{filter}}, T_{\text{filter}})$, diagonal Fisher information matrix $\mathcal{I}$

1: HI = indicies of $T_{\text{head}}$ most important heads in each layer
2: FI = indicies of $T_{\text{filter}}$ most important filters in each layer
3: $H' = H - T_{\text{head}}$
4: $N' = N - T_{\text{filter}}$
5: $C' = C - L(c_{\text{head}} + c_{\text{filter}})$
6: **for** $n = 0$ **to** $LH'$ **do**
7: $\quad I$ = indicies of $n$ most important heads not in HI
8: $\quad f = \lfloor (C' - na_{\text{head}})/a_{\text{filter}} \rfloor$
9: $\quad J$ = indicies of $f$ most important filters not in FI
10: $\quad \text{HI}' = \text{HI} \cup I$
11: $\quad \text{FI}' = \text{FI} \cup J$
12: $\quad S[n] = \sum_{i \in \text{HI}' \cup \text{FI}'} \mathcal{I}_{ii}$
13: $\quad R[n] = (\text{HI}', \text{FI}')$
14: **end for**
15: $n^* = \arg \max_n S[n]$
16: $\text{HI}^*, \text{FI}^* = R[n^*]$
17: Initialize $\text{m}^{\text{MHA}}$ and $\text{m}^{\text{FFN}}$ as 0
18: $\text{m}^{\text{MHA}}[\text{HI}^*] = 1$
19: $\text{m}^{\text{FFN}}[\text{FI}^*] = 1$
**Output:** $\text{m}^* = (\text{m}^{\text{MHA}}, \text{m}^{\text{FFN}})$

## A.3 Derivation of Equation 11

Based on Eq. 5 and the warm-start constraint in Section 4.2, the optimization problem Eq. 4 is written as follows:

$$\arg \min_{\text{m}} (\mathbb{1} - \text{m})^{\mathsf{T}} \mathcal{I} (\mathbb{1} - \text{m}), \tag{20}$$

$$\text{s.t. } ||\text{m}_l||_0 = ||\text{m}_l^*||_0 \text{ for } l = 1, 2, \ldots, L, \tag{21}$$

where $\text{m}^*$ is the mask searched in Section 4.1 using the diagonal approximation of $\mathcal{I}$. Under the block diagonal assumption, Eq. 20 can be reduced as follows:

$$\arg \min_{\text{m}} (\mathbb{1} - \text{m})^{\mathsf{T}} \mathcal{I} (\mathbb{1} - \text{m}) \approx \arg \min_{\text{m}} \sum_{l=1}^{L} (\mathbb{1} - \text{m}_l)^{\mathsf{T}} \mathcal{I}_l (\mathbb{1} - \text{m}_l), \tag{22}$$

where $\mathcal{I}_l$ is the $l$-th diagonal block of $\mathcal{I}$. Here what we want to show is that the problem Eq. 22 can be solved by independently solving the optimization problem Eq. 11 for each layer. We prove this by contradiction. Suppose that $\hat{\text{m}} = (\hat{\text{m}}_1, \ldots, \hat{\text{m}}_L)$ is the mask obtained by solving Eq. 11 for each layer. If there exists a mask m that strictly better optimizes Eq. 22 than $\hat{\text{m}}$:

$$\sum_{l=1}^{L} (\mathbb{1} - \text{m}_l)^{\mathsf{T}} \mathcal{I}_l (\mathbb{1} - \text{m}_l) < \sum_{l=1}^{L} (\mathbb{1} - \hat{\text{m}}_l)^{\mathsf{T}} \mathcal{I}_l (\mathbb{1} - \hat{\text{m}}_l), \tag{23}$$

while also satisfying the constraint Eq. 21, then there must exist a layer $k$ such that

$$(\mathbb{1} - \text{m}_k)^{\mathsf{T}} \mathcal{I}_k (\mathbb{1} - \text{m}_k) < (\mathbb{1} - \hat{\text{m}}_k)^{\mathsf{T}} \mathcal{I}_k (\mathbb{1} - \hat{\text{m}}_k). \tag{24}$$

However, this contradicts the assumption that $\hat{\text{m}}_k$ is the minimizer of Eq. 11 for layer $k$. Therefore, such a mask as m cannot exist, and $\hat{\text{m}}_k$ is the optimal solution for Eq. 22.

## A.4 Formulating Equation 12 as a Linear Least Squares Problem

For a MHA layer, the problem of minimizing reconstruction error can be written as follows:

$$\arg \min_{\text{m}_l^{\text{MHA}}} ||(\text{x} + \text{MHA}(\text{x}; \text{m}_l^{\text{MHA}})) - (\text{x}' + \text{MHA}(\text{x}'; \mathbb{1}))||_2^2, \tag{25}$$

$$\text{s.t. } Z(\text{m}_l^{\text{MHA}}) = Z(\hat{\text{m}}_l^{\text{MHA}}), \tag{26}$$

where $\hat{\text{m}}$ is the mask obtained as the result of mask rearrangement (Section 4.2) and $Z(\text{m})$ denotes the indices of zero entries in m. Eq. 26 is the constraint we impose in Section 4.3 that the zero-valued

mask variables in m̂ are fixed to 0 so that the tuned mask also satisfies the FLOPs/latency constraint. Then we rewrite the problem as the following linear least squares problem:

$$\underset{\mathrm{m}_l}{\arg\min} \, ||\mathrm{A}\mathrm{m}_l - \mathrm{b}||_2^2, \tag{27}$$

$$\text{where } \mathrm{A} := [\hat{m}_{l,1}^{\mathrm{MHA}}\mathrm{Att}_1(\mathrm{x}), \ldots, \hat{m}_{l,H}^{\mathrm{MHA}}\mathrm{Att}_H(\mathrm{x})] \text{ and } \mathrm{b} := \Big(\mathrm{x}' + \sum_{h=1}^{H} \mathrm{Att}_h(\mathrm{x}')\Big) - \mathrm{x}. \tag{28}$$

Here, the elements of $\hat{m}_l^{\mathrm{MHA}}$ are multiplied to the matrix A to ensure that the output activations of the pruned heads are not used to reconstruct the original output. Although Eq. 27 has a closed form solution $(\mathrm{A}^\mathsf{T}\mathrm{A})^{-1}\mathrm{A}^\mathsf{T}\mathrm{B}$, we use the numerical solver in CuPy for higher stability.

## A.5 Experimental Details

### A.5.1 Experimental Setup

Our framework is implemented on top of PyTorch v1.9.1 [57] and HuggingFace Transformers v4.12.0 [86]. For the baseline, we downloaded the pre-trained checkpoints from the HuggingFace Transformers repository, and we fine-tuned them on GLUE [78] and SQuAD [61, 60] datasets with the standard training recipe. We then use 2K examples from the training sets to prune the baseline models. We report accuracy for GLUE tasks, except for STS-B that we report Spearman Correlation, and F1 score for SQuAD tasks on the development sets. All experiments in this paper are conducted on an AWS p3.2xlarge instance which has an NVIDIA V100 GPU. We used seed numbers from 0 to 9, and we reported the averaged results.

### A.5.2 Datasets

GLUE tasks [78] include sentence similarity (QQP [30], MRPC [13], STS-B [5]), sentiment classification (SST-2 [69]), textual entailment (RTE [11]) and natural language inference (MNLI [85], QNLI [61]). There are 364K, 4K, 6K, 67K, 3K, 392K, 105K training examples, respectively. We exclude CoLA [84] and WLNI [42] due to their unstable behaviors.

SQuAD 1.1 [61] and SQuAD 2.0 [60] are question and answering tasks, each of which contains 88K and 130K training examples. SQuAD 2.0 is an extension of SQuAD 1.1 by including unanswerable questions whose answers are not stated in the given contexts.

## A.6 Impact of Sample Dataset Size

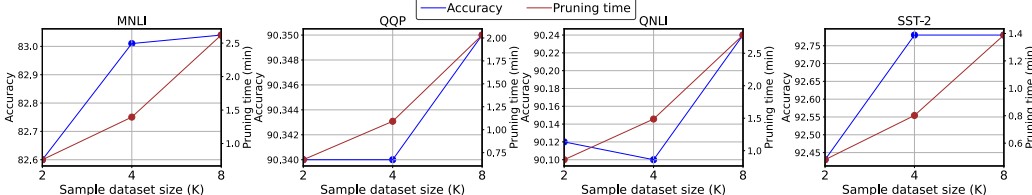

Figure 8: Accuracy and pruning time with 2K, 4K, 8K samples. The FLOPs constraint is 60%.

There is a trade-off between sample dataset size and accuracy. Figure 8 shows the correlation between sample size, pruning time, and accuracy for 4 GLUE datasets (with more than 64K training examples). Note that for simplicity we used 2K samples in all our experiments in Section 5. Figure 8 demonstrates that using more examples can improve our accuracy results by up to 0.4% with 2–4× longer pruning time (which is still less than 3 minutes).

## A.7 Details for the Comparison with the Prior Methods

We compare the FLOPs-accuracy trade-off of our method to the prior structured pruning methods for Transformers in Flop [83], SLIP [44], Sajjad et al. [62], DynaBERT [25], EBERT [49], and Block Movement Pruning (BMP) [38] on 4 GLUE tasks, QQP, QNLI, SST-2, and MRPC. We use the FLOPs

Table 4: The absolute accuracy and the amount of accuracy degradation from the baseline after pruning BERT$_{BASE}$ using our method and the prior structured pruning methods with different relative FLOPs. $^{\dagger}$Reported as F1 score instead of accuracy.

| Method | Rel. FLOPs | Accuracy | | | | Accuracy Diff | | | |
|---|---|---|---|---|---|---|---|---|---|
| | | QQP | QNLI | SST-2 | MRPC | QQP | QNLI | SST-2 | MRPC |
| Flop [83] | Baseline | - | 91.6 | 92.7 | 90.9$^{\dagger}$ | - | 0 | 0 | 0 |
| | 66.7 | - | 89.0 | 92.1 | 88.6$^{\dagger}$ | - | -2.6 | -0.6 | -2.3 |
| SLIP [44] | Baseline | 90.6 | 91.6 | 92.7 | 90.9$^{\dagger}$ | 0 | 0 | 0 | 0 |
| | 65.6 | 89.7 | 90.7 | 91.7 | 89.9$^{\dagger}$ | -0.9 | -0.9 | -1.0 | -1.0 |
| | 61.5 | 88.9 | 89.5 | 91.8 | 88.1$^{\dagger}$ | -1.7 | -2.1 | -0.9 | -2.8 |
| Sajjad et al. [62] | Baseline | 91.1 | 91.1 | 92.4 | 88.0 | 0 | 0 | 0 | 0 |
| | 66.7 | 90.6 | 89.7 | 90.6 | 79.4 | -0.4 | -1.4 | -1.8 | -8.6 |
| | 50.0 | 90.4 | 87.6 | 90.3 | 80.2 | -0.7 | -3.5 | -2.2 | -7.8 |
| DynaBERT [25] | Baseline | - | - | 92.9 | 87.7 | - | - | 0 | 0 |
| | 75.0 | - | - | 92.3 | 86.0 | - | - | -0.6 | -1.7 |
| | 50.0 | - | - | 91.9 | 86.0 | - | - | -1.0 | -1.7 |
| EBERT [49] | Baseline | 87.9 | 91.5 | 93.2 | - | 0 | 0 | 0 | - |
| | 60.0 | 87.5 | 90.2 | 92.2 | - | -0.4 | -1.3 | -1.0 | - |
| | 50.0 | 87.0 | 89.6 | 91.6 | - | -0.7 | -1.9 | -1.6 | - |
| BMP [38] | Baseline | 91.1 | - | 92.7 | - | 0 | - | 0 | - |
| | 50.0 | 90.4 | - | 90.7 | - | -0.7 | - | -2.0 | - |
| BMP [38] (Reproduced) | Baseline | - | - | 92.7 | - | - | - | 0 | - |
| | 72.6 | - | - | 92.1 | - | - | - | -0.6 | - |
| | 56.5 | - | - | 91.5 | - | - | - | -1.2 | - |
| | 46.7 | - | - | 90.8 | - | - | - | -1.9 | - |
| Ours | Baseline | 91.0 | 91.4 | 93.6 | 86.3 | 0 | 0 | 0 | 0 |
| | 70.0 | 90.7 | 90.9 | 93.0 | 86.1 | -0.3 | -0.5 | -0.6 | -0.2 |
| | 60.0 | 90.4 | 90.0 | 92.5 | 85.3 | -0.6 | -1.4 | -1.1 | -1.0 |
| | 50.0 | 89.5 | 88.7 | 91.6 | 83.2 | -1.5 | -2.7 | -2.0 | -3.1 |

and accuracy values reported in each paper (for Flop, we use the values reported in SLIP). To make a fair comparison, we use the experimental results *without* any additional knowledge distillation or data augmentation in each paper. Because all of these papers have slight differences in their baseline accuracy, it is difficult to directly compare the absolute accuracy for the pruned models. Therefore, we use the amount of the accuracy drop (i.e., accuracy of the pruned model subtracted by the accuracy of the original model) instead. For Flop and SLIP, the results for MRPC are reported as F1 score instead of accuracy. For these cases, we report that the amount of the F1 score drop instead, assuming that it is similar to the amount of the accuracy drop. Since BMP only reports the parameter counts reduction, we directly use this value as FLOPs reduction, under an assumption that the head pruning ratio is similar to the filter pruning ratio. We include the full table in Table 4.

## A.8 Performance at high sparsity

For high sparsity, our framework can be used with retraining to recover the accuracy. In this experiment, our framework skips Mask Tuning (Section 4.3) and retrains the model parameters with the binary mask fixed. Specifically, it retrains the pruned BERT$_{BASE}$ model for 3 epochs with the full training dataset with learning rate $2e^{-5}$. The retraining cost can be considered the same as that of Sajjad et al. and much lower than DynaBERT [25], EBERT [49], and BMP [38] (more details in Section A.9).

Figure 9 shows the FLOPs-accuracy comparison between our method and the prior structured pruning methods at high sparsity. For fair comparison, we used the results *without* knowledge distillation and data augmentation reported in each paper. For the SST-2 dataset, our method performs comparably to DynaBERT and outperforms all other methods. For the MRPC dataset, our framework consistently outperforms all of the baselines. These results imply that our Mask Search (Section 4.1) and Mask

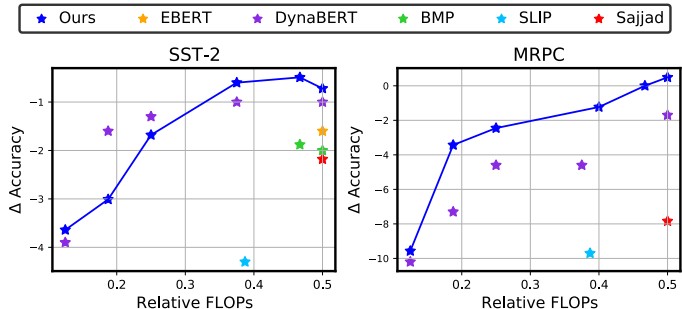

Figure 9: Performance comparison at high sparsity. Each method (including ours) *retrains* the pruned models *without* data augmentation and knowledge distillation.

Rearrangement (Section 4.2) find more optimal binary masks than other methods even at high sparsity, which is consistent with Figure 7.

## A.9 Retraining Costs

DynaBERT [25] consists of 2-stage training, 1 epoch of width and depth-adaptive training followed by additional 3 epochs of final fine-tuning. For the first stage, it jointly trains 12 different width and depth configurations, roughly adding up $12\times$ training costs than the normal training. EBERT [49] consists of 2-stage training, 3 epochs that jointly trains the pruning parameters and the model weights, followed by 3 additional epochs of final fine-tuning. BMP [38] requires 20 epochs of training to search for the optimal pruning configuration and retrain the model weights. Similarly, CoFi [88] requires 20 epochs of pruning and 20 epochs of post-pruning fine-tuning. We measure end-to-end retraining latency on an NVIDIA V100 GPU using a batch size of 32 for all experiments.

## A.10 Pruning Time Breakdown

Table 5: Time breakdown (in percentage) of our pruning framework on a single NVIDIA V100 GPU. It consists of Gradient Computation (GC), Mask Search (MS, Section 4.1), Mask Rearrangement (MR, Section 4.2), and Mask Tuning (MT, Section 4.3). In the last column, we provide the total amount of time for end-to-end pruning, in seconds.

|  | GC | MS | MR | MT | Total (s) |
|---|---|---|---|---|---|
| GLUE | 23.8% | 0.3% | 9.4% | 66.5% | 39.3 |
| SQuAD | 20.5% | 0.1% | 3.5% | 76.0% | 135.1 |

## A.11 Efficacy of the Fisher-based Importance Metric

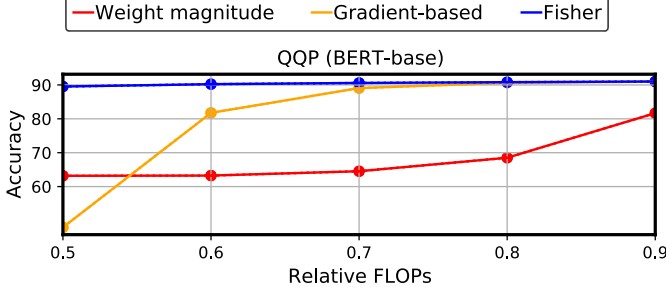

Figure 10: Pruning performance of our pipeline when different importance metrics are used. Note that the mask re-arrangement stage is *skipped* in this experiment.

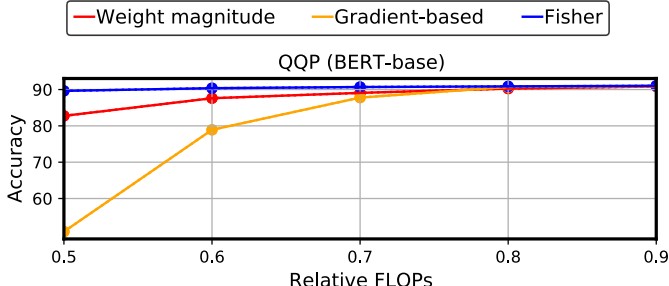

Figure 11: Pruning performance of our pipeline when different importance metrics are used. Note that the mask re-arrangement stage is *included* in this experiment.

To demonstrate the efficacy of the Fisher-based importance score, we compare its performance with two other importance metrics (i.e., weight magnitude and gradient-based), which can be plugged into the mask search algorithm. However, since the mask re-arrangement algorithm needs signals that capture the interaction between mask variables, the mask re-arrangement technique can only be used with the Fisher information matrix. Hence, we designed two experiments: without and with the mask re-arrangement stage. Figure 10 shows the results when the mask re-arrangement is skipped. Figure 11 shows the results when the Fisher-based mask re-arrangement is applied to all. In both experiments, our Fisher importance metric significantly outperforms the two other metrics.

### A.12 Efficacy of Mask Search and Mask Re-arrangement

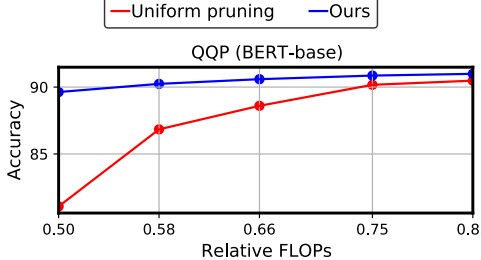

Figure 12: Performance of our method and Fisher-based uniform pruning. Mask tuning is applied to both methods.

To show the effectiveness of the mask search and re-arrangement stages, we compare the performance of our method with uniform Fisher pruning, which prunes every layer with the same sparsity. Mask tuning is applied to both methods. Figure 12 shows that the accuracy of our method significantly outperforms that of uniform pruning by up to 8.6%. The result demonstrates the necessity of our mask search and re-arrangement techniques in finding quality binary masks.

### A.13 Societal Impacts

We believe our work would not bring immediate negative impacts on the society, as it aims to accelerate the model inference without affecting the output quality. Our work can partly relieve the environmental concern due to DNN training, as it eliminates the need of re-training after pruning.