# OpenReview forum: "A Fast Post-Training Pruning Framework for Transformers"
_NeurIPS.cc/2022/Conference — NeurIPS 2022 Accept_

### Official Review · Reviewer_Lp5s · 2022-06-27

**Rating:** 6
**Confidence:** 4
**Soundness:** 3 good
**Presentation:** 2 fair
**Contribution:** 2 fair

**Summary:**

This paper proposes a three-stage post-training pruning framework for Transformers.  It first uses Fisher information to search the binary mask, i.e, layer-wise pruning rate. Then, the framework modifies the binary mask in a layer-wise manner. Lastly, it tunes the non-zero valued mask for minimizing the layer-wise reconstruction error.

Such a framework could retain the performance of the model without retraining, thus it can finish the pruning in less than 3 minutes on a single GPU and can obtain actual speedup in inference because of structured pruning.

Extensive experiments show that the proposed post-training pruning framework has a comparable performance with prior methods.


**Questions:**

1. Figure 3 (Left) only shows that  the actual speed is linear to the number of filters, which does not support the claim in line206-207. I suggest authors re-plot Figure 3 (Left) with fewer filter number.

2. Technical merits are limited.  Each component used in the three-stage pruning has been used in prior works. For example,   taking Fisher Information Matrix as an importance score and  tuning the mask variables toward minimizing the layer-wise reconstruction error.

3. I can not understand why the mask by Eq. (4) and Eq. (11) is different because the importance score is the same.   Line 237 claims that $||m_l||_0$ is  equal to $||m_l^*||$. Line 242 claims that $\mathcal{I}_l$ is the  $l$-th diagonal block of $\mathcal{I}$. In such situation, $m_l^*$ selected by $\mathcal{I}$  and $m_l$ selected by $\mathcal{I}_l$ should be the same. This can be proved by contradiction.

4. The ablation experiments are not clear enough. There are two ablative modules in Figure 6, including importance score and mask search algorithm. Such an ablation study with one more ablative module can not support the claim of Line 319.  The ablation experiment in Figure 7 should be divided into two experiments. 1) The importance score. Keeping the three-stage pruning pipeline and just modifying the importance score, which can reflect the efficacy of the importance score. 2) Mask search and rearrangement. Keeping the importance score and using uniformly prune, can reflect the efficacy of mask search and re-arrangement.


typos:

- Line 613: Eq. ?? -> Eq. 9
- Algorithm 2.2 pf->of

==================post rebuttal============
My concerns are addressed. I raise up my score. Thanks.

**Limitations:**

No societal impact discussion needed, in my opinion.

**Strengths And Weaknesses:**

1. To my knowledge, this is the first work to use post-training pruning in Transformer. I recognize the contribution of applying technologies to new areas.

2. Proposing simple mask search solutions based on FLOPs and Latency,  which avoids user intervention.

3. Experiments show the efficacy  of the proposed framework, which could retain high accuracy without retraining.

---

> ### Author Response · Authors · 2022-08-02
> **Author Response to Reviewer Lp5s [Question 4]**
>
> > **[Question4-1]** The ablation experiments are not clear enough. There are two ablative modules in Figure 6, including importance score and mask search algorithm. Such an ablation study with one more ablative module can not support the claim of Line 319. The ablation experiment in Figure 7 should be divided into two experiments. 1) The importance score. Keeping the three-stage pruning pipeline and just modifying the importance score, which can reflect the efficacy of the importance score.
>
> In Appendix A.12, we conduct experiments to demonstrate the efficacy of our Fisher-based importance score. We compare the performance when the three different importance metrics (i.e., weight magnitude, gradient-based, Fisher) are plugged into our pruning pipeline. However, we note that while our mask search algorithm can be used with any importance metric, the mask re-arrangement technique requires the Fisher information matrix because the algorithm requires signals that capture the interaction between mask variables. Hence, we designed the following two ablation experiments. First, we skip the mask re-arrangement stage for all importance metrics (Figure 9, added). Second, we include the Fisher-based mask re-arrangement stage for all importance metrics (Figure 10, added). In both experiments, our Fisher-based importance score consistently leads to the highest accuracy.
>
> > **[Question4-2]** 2) Mask search and rearrangement. Keeping the importance score and using uniformly prune, can reflect the efficacy of mask search and re-arrangement.
>
> To show the effectiveness of mask search and re-arrangement, we compare its performance with uniform Fisher pruning, which prunes every layer with the same sparsity level. Mask tuning is applied to both methods. Figure 11 in Appendix A.13 shows that the accuracy of our method significantly outperforms that of uniform pruning by up to 8.6\%. The result demonstrates the necessity of our mask search and re-arrangement techniques in finding quality binary masks.
>
> **[Typos]** We thank the reviewer for letting us know the typos. They are fixed in the updated paper.

---

> ### Author Response · Authors · 2022-08-02
> **Author Response to Reviewer Lp5s [Question 1, 2, 3]**
>
> Thank you for your detailed review and suggestions. We have added the new ablation experiments in our paper. Here we address your questions in detail:
>
> > **[Question1]** Figure 3 (Left) only shows that the actual speed is linear to the number of filters, which does not support the claim in line206-207. I suggest authors re-plot Figure 3 (Left) with fewer filter number.
>
> That is an excellent point. We have re-plotted Figure 3 with fewer number of filters so that the non-linearity of LAT is more clearly shown. The threshold numbers of filters that the latency starts increasing are 1536, 672, 288 for the batch size 8 (blue line), 16 (green line), 32 (yellow line), respectively.
>
> > **[Question2]** Technical merits are limited. Each component used in the three-stage pruning has been used in prior works. For example, taking Fisher Information Matrix as an importance score and tuning the mask variables toward minimizing the layer-wise reconstruction error.
>
> While the individual methods  might not be regarded as new in isolation, we are the first attempt to bring and carefully adjust these ideas to the re-training free setting of Transformers. For instance, using the Fisher information matrix as a sensitivity matrix to guide pruning (or other model compression methods) has been explored in prior literature; however, our mask search and re-arrangement are a novel approach to find an accurate pruning pattern under a *block*-diagonal assumption of the Fisher information matrix by bringing down the approximation problem into a tractable scale. This was particularly critical for the re-training free setting where (i) the pruning pattern should be accurate enough to retain critical information without retraining and (ii) the search time should be fast. Taken together, our result proves the potential and efficacy of re-training free pruning for Transformer architectures for the first time, which we expect to have a substantial academic and industrial impact.
>
> > **[Question3]** I can not understand why the mask by Eq. (4) and Eq. (11) is different because the importance score is the same. Line 237 claims that $||m_l||_0$ is equal to $||m_l^\ast||$. Line 242 claims that $\mathcal{I}_l$ is the $l$-th diagonal block of $\mathcal{I}$. In such situation, $m_l^\ast$ selected by $\mathcal{I}$ and $m_l$ selected by $\mathcal{I}_l$ should be the same. This can be proved by contradiction.
>
> Since the Fisher information matrix is not in fact diagonal nor block-diagonal, the masks obtained from the diagonal/block-diagonal/full Fisher information matrix can be different. Below we provide an example to illustrate this. Assume that we have a two-layer NN where each layer has 3 filters (i.e., 6 filters in total), and we have to prune 3 out of the 6 filters. Say the full Fisher information matrix $\mathcal{I}$ is as follows:
>
> $\mathcal{I} =
> \begin{pmatrix}
> 0 & -1 & 2 & 0 & 0 & 1 \\\\
> 0 & 4 & -3 & 1 & 0 & 0 \\\\
> 1 & 2 & 5 & 0 & -1 & 0 \\\\
> 0 & 0 & 0 & 2 & 5 & 3 \\\\
> 0 & 1 & 0 & 1 & 1 & -7 \\\\
> -1 & 0 & 0 & 4 & -4 & 3 \\\\
> \end{pmatrix}$
>
> Under the diagonal assumption, we can first obtain the warmup mask $\text{m}^{\ast}$ as $(m_1, m_2, m_3, m_4, m_5, m_6) = (0, 1, 1, 0, 0, 1)$ using Algorithm 1. Then, by the block-diagonal approximation, we decompose $\mathcal{I}$  into two sub-matrices $\mathcal{I}_1$ and $\mathcal{I}_2$:
>
> $\mathcal{I}_1 =
> \begin{pmatrix}
> 0 & -1 & 2 \\\\
> 0 & 4 & -3 \\\\
> 1 & 2 & 5 \\\\
> \end{pmatrix}, \quad
> \mathcal{I}_2 =
> \begin{pmatrix}
> 2 & 5 & 3 \\\\
> 1 & 1 & -7 \\\\
> 4 & -4 & 3 \\\\
> \end{pmatrix}$
>
> By the warmup constraint, $|m_1| + |m_2| + |m_3| = 2$ and $|m_4| + |m_5| + |m_6| = 1$. Here, the optimal mask for $\mathcal{I}_1$ is $(m_1, m_2, m_3) = (0, 1, 1)$ (i.e., the mask variables are not re-arranged). However, as the non-diagonal elements are considered, the optimal mask for $\mathcal{I}_2$ changes to $(m_4, m_5, m_6) = (1, 0, 0)$. Thus, the re-arranged mask $\hat{\text{m}} = (0, 1, 1, 1, 0, 0)$ is different from the warmup mask $\text{m}^\ast = (0, 1, 1, 0, 0, 1)$.

---

> ### Comment · Reviewer_Lp5s · 2022-08-09
> **rebuttal decision from Reviewer Lp5s**
>
> ==================post rebuttal============
>
> My concerns are addressed. I raise up my score. Thanks.

---

### Official Review · Reviewer_hRDe · 2022-07-10

**Rating:** 6
**Confidence:** 4
**Soundness:** 3 good
**Presentation:** 4 excellent
**Contribution:** 3 good

**Summary:**

In this paper, the authors proposed a fast post-training pruning framework for transformer-based language models which does not require retraining. The algorithm takes a model, sample dataset, and compression constraint to generate the compressed model. It introduces three techniques to retain high accuracy: 1. mask search; 2. mask rearrangement; 3. mask tuning. Experiments show that the proposed method achieves 2x FLOPs reduction and 1.6x speed up within 1% accuracy drop.

**Questions:**

1. The authors may discuss if the proposed method is only restricted to transformers. Can it be applied to other models like CNNs? Is it possible that the post-training method only works well here due to the large redundancy of BERT models on a specific downstream dataset?


**Limitations:**

There lacks a discussion of potential negative societal impacts. It might be acceptable due to the nature of the work; but a discussion is still encouraged.

**Strengths And Weaknesses:**

Strengths:
1. Firstly, the paper is well written and easy to follow.
2. The proposed method solves a complex optimization problem by introducing several approximations and using a multi-step approach. The process is introduced clearly. The effectiveness of each step is demonstrated by ablation studies.
3. The experimental results are solid. The proposed method achieves similar performance compared to existing work without a large training cost, which could be valuable to real-life applications.
4. I like the discussion of latency-aware compression, where the authors used a piece-wise linear function to approximate the latency LUT, which is integrated into the optimization objective. It is a smart design to fit both settings under the same optimization framework.

Weakness:
1. Firstly, it seems that the proposed method is not limited to pruning transformers; it can be also applied to other models like CNNs by just using the channel mask. Is there any consideration why the work is limited to transformers? Will the algorithm perform well on other models like CNNs?
2. There are some other works on post-training channel pruning (e.g., [a]). It would be better if the authors can show the proposed method can also outperform general data-free pruning methods when applied to transformers. But I also agree the current experimental results are already solid.

[a] Lazarevich et al., Post-training deep neural network pruning via layer-wise calibration, ICCVW.

---

> ### Author Response · Authors · 2022-08-02
> **Author Response to Reviewer hRDe**
>
> Thank you for your review and thoughtful comments. Here we address your questions:
>
> > **[Weakness1]** Firstly, it seems that the proposed method is not limited to pruning transformers; it can be also applied to other models like CNNs by just using the channel mask. Is there any consideration why the work is limited to transformers? Will the algorithm perform well on other models like CNNs?
>
> > **[Question1]** The authors may discuss if the proposed method is only restricted to transformers. Can it be applied to other models like CNNs? Is it possible that the post-training method only works well here due to the large redundancy of BERT models on a specific downstream dataset?
>
> In this paper, we focused on Transformer pruning because the paper was motivated by our initial observation that existing post-training CNN pruning methods cannot be applied to Transformers, which we elaborate in [Weakness2]. However, we think our method can be extended to CNNs as well with a minor modification in Algorithm 1. The current version of Algorithm 1 leverages the fact that the input tensor shape is the same across every layer in a Transformer, which is not the case for CNNs. With a further consideration for this difference, our method can be extended to CNNs, which we will clarify in the final version of the paper.
>
> Moreover, we acknowledge the reviewer's concern that our post-training pruning method might have benefited from the large redundancy of BERT models; however, we would also like to highlight that our framework was effective for  pruning DistilBERT, which is already a compressed model with significantly less redundancy.
> Furthermore, our extensive experiments over 8 different tasks (including classification, regression, and question answering) demonstrate that our framework can consistently achieve good performance across different tasks as well.
>
> > **[Weakness2]** There are some other works on post-training channel pruning (e.g., [a]). It would be better if the authors can show the proposed method can also outperform general data-free pruning methods when applied to transformers. But I also agree the current experimental results are already solid.
>
> While there exist post-training pruning methods for CNNs, we find it difficult to extend those techniques to Transformer pruning because their underlying ideas are often tightly coupled with the architectural characteristics of CNNs. For example, Neuron Merging [1] exploits the equation $\text{ReLU}(ax) = a\text{ReLU}(x)$ if $a \geq 0$, which does not hold for GELU. For another example, RED [2] requires a model to be a repeating structure of linear layers and element-wise activations, and thus cannot be applied to MHA layers.
>
> The proposed method in [3] ([a] in the reviewer's comment) can be applied to Transformer pruning. However, we did not consider it as our baseline or competitor method as it is an unstructured pruning method whereas our main focus in the paper is on structured pruning. Moreover, we show in Figure 6 that magnitude pruning of Transformers as in [3] leads to significant accuracy drop without re-training.
>
> [1] Kim, Woojeong, et al. "Neuron merging: Compensating for pruned neurons." Advances in Neural Information Processing Systems 33 (2020): 585-595.
>
> [2] Yvinec, Edouard, et al. "RED: Looking for Redundancies for Data-FreeStructured Compression of Deep Neural Networks." Advances in Neural Information Processing Systems 34 (2021): 20863-20873.
>
> [3] Lazarevich, Ivan, Alexander Kozlov, and Nikita Malinin. "Post-training deep neural network pruning via layer-wise calibration." 2021 IEEE/CVF International Conference on Computer Vision Workshops (ICCVW). IEEE, 2021.
>
> > **[Societal Impacts]** There lacks a discussion of potential negative societal impacts. It might be acceptable due to the nature of the work; but a discussion is still encouraged.
>
> Thank you for your suggestion. We have added a brief discussion on the potential societal impacts of this paper in Appendix A.14. Due to the space limit, we did not include the section in the main paper.

---

> > ### Comment · Reviewer_hRDe · 2022-08-07
> > **Review Update**
> >
> > Thanks for providing the feedback. The rebuttal addresses my concerns. I would like to keep my acceptance recommendation.

---

### Official Review · Reviewer_CMhk · 2022-07-11

**Rating:** 7
**Confidence:** 4
**Soundness:** 3 good
**Presentation:** 3 good
**Contribution:** 3 good

**Summary:**

This paper proposes three  techniques to obtain a high accuracy transformer without  retraining. A search algorithm to find which heads and filters are need to prune based on the Fisher information. An algorithm which rearrange the mask that complements the search algorithm. And tuning the mask which reconstructs the output activations for each layer. The experiments show the author can get better results than compare methods.

**Questions:**

1、How to obtain latency exactly? Is the latency become different when obtain them from different hardware and inference engines



**Limitations:**

Please see my weaknesses

**Strengths And Weaknesses:**

Strengths:

This paper is well-written, well-motivated, and clear presentation.

The proposed algorithm improves transformer throughout efficiency with competitive accuracy and small latency, outperforming prior pruning and distillation approaches.

Formulating the hardware-aware structural pruning as a knapsack problem is interesting.

Weaknesses:

This paper should compare their algorithms with some SOTA pruning methods on transformer such as COFI[1]

[1]Xia M, Zhong Z, Chen D. Structured pruning learns compact and accurate models[J]. arXiv preprint arXiv:2204.00408, 2022.

---

> ### Author Response · Authors · 2022-08-02
> **Author Response to Reviewer CMhk**
>
> Thank you for your review and positive feedback. We have added performance comparison with CoFi in the updated version of the paper. Here we address your comments in detail:
>
> > **[Weakness1]** This paper should compare their algorithms with some SOTA pruning methods on transformer such as COFI.
>
> We thank the reviewer for introducing the new related work. In Figure 5, we have added the performance of CoFi on SST-2 and MRPC datasets without knowledge distillation and data augmentation.
> We will add more data points including the results on the QQP and QNLI datasets in the final version of the paper.
>
> The results align with our previous experiments in which our framework exhibits comparable or better accuracy than the prior works even without re-training of the pruned model. On the MRPC dataset, CoFi marginally outperforms our results. However, on the SST-2 dataset, CoFi shows larger accuracy drop than ours. We would also like to note that CoFi requires at least 40 epochs for pruning and re-training, which amounts to **7 GPU hours** for SST-2. In contrast, the end-to-end time of our pruning method is **only 39 seconds**. Pruning time comparison on the MNLI dataset is also provided in Table 1.
>
> > **[Question1]** How to obtain latency exactly? Is the latency become different when obtain them from different hardware and inference engines.
>
> For latency objective pruning, our framework takes a latency lookup table as an input. The lookup table can be obtained by measuring the MHA/FFN layer latencies with different numbers of heads/filters, using the target hardware and inference engine. In this way, our framework adapts to diverse hardware and software backends. For the results in Table 3 in Appendix, we used PyTorch with a V100 GPU to generate the lookup table and to measure the latencies of the pruned models.

---

### Official Review · Reviewer_92Qy · 2022-07-11

**Rating:** 6
**Confidence:** 4
**Soundness:** 3 good
**Presentation:** 4 excellent
**Contribution:** 3 good

**Summary:**

The paper proposed a post-training pruning framework for Transformers, without retraining. It prunes of both heads in MHA and filters in FFN layers in a structured  way. The process is done by applying a lightweight Fisher based mask search along with a Fisher mask rearrangement and mask tuning. The results are comparable or even better FLOPs-accuracy trade-off than prior methods.

**Questions:**

1. The reason to use second-order Taylor. It seems that first-order Taylor can be faster and doesn't lose much accuracy? Some other works use first-order [1][2]


[1]Accelerating Sparse DNN Models without hardware-Support via Tile-Wise Sparsity
[2]Chasing sparsity in vision transformers: an end-to-end exploration

**Limitations:**

Please see the above comments. I will consider changing my ratings based on the author's rebuttal.



**Strengths And Weaknesses:**

Strengths: The paper is well written, and the authors proposed the method in quite a detail. The post-training pruning framework does not require retraining, which is very good. Adding latency-constrained in consideration is also good.
The experiments are quite sufficient and are able to support their claims and conclusions. The results show the effectiveness of the proposed methods. The paper also compared existing structured pruning works on GLUE

---

> ### Author Response · Authors · 2022-08-02
> **Author Response to Reviewer 92Qy**
>
> Thank you for your review and positive feedback. Here we address your question:
>
> > **[Question1]** The reason to use second-order Taylor. It seems that first-order Taylor can be
> faster and doesn’t lose much accuracy? Some other works use first-order.
>
> This is a good question. Many of the previous works used first-order Taylor expansion and ignored the second order term, mostly because of its computational cost. However, the Hessian matrix can be efficiently approximated by Fisher information matrix, which can be computed using gradients. Thus, our Fisher-based importance score can be considered as an efficient and more accurate substitute for first-order based methods. Our empirical results and end-to-end timings also support this.

---

> > ### Comment · Reviewer_92Qy · 2022-08-09
> > **Post-rebuttal**
> >
> > I thank the author for the rebuttal. The author solves my concern.

---

### Meta-Review · Area_Chair_XTL6 · 2022-08-28

**Recommendation:** Accept
**Confidence:** Certain

**Metareview:**

The authors deliver on what they promise: a fast post-training pruning framework for transformers. It reduces the inference costs of deploying transformers while preserving much or all of their accuracy on the standard range of academic downstream tasks. Moreover, it does so without the hefty costs that typically come with prune-and-retrain cycles. The paper is clearly written and well-presented, and the technique seems to work quite well. The authors seemed to satisfactorily address all reviewer concerns, and those concerns were minor at best. What more can you ask for? I look forward to visiting the poster at NeurIPS and trying this technique myself.

The authors are to be especially commended for focusing on real-world speedup on real hardware. That's (sadly) still a rarity in pruning papers. This is something that appears genuinely useful, today, by practitioners.

**Award:**

No

---

### Decision · Program_Chairs · 2022-09-14

Accept